# Rangeland dynamics: investigating vegetation composition and structure of urban and exurban prairie dog habitat

Rebecca Hopson[1], Paul Meiman[1] and Graeme Shannon[2]

[1] Department of Forest and Rangeland Stewardship, Colorado State University, Fort Collins, CO, USA
[2] Department of Fish, Wildlife, and Conservation Biology, Colorado State University, Fort Collins, CO, USA

## ABSTRACT

Rapid human population growth and habitat modification in the western United States has led to the formation of urban and exurban rangelands. Many of these rangelands are also home to populations of black-tailed prairie dogs (*Cynomys ludovicianus*). Our study aimed to compare the vegetation composition of an urban and exurban rangeland, and explore the role that prairie dogs play in these systems. The percent absolute canopy cover of graminoids (grasses and grass-likes), forbs, shrubs, litter, and bare ground were estimated at sampling areas located on and off prairie dog colonies at an urban and an exurban site. Herbaceous forage quality and quantity were determined on plant material collected from exclosure cages located on the colony during the entire growing season, while a relative estimate of prairie dog density was calculated using maximum counts. The exurban site had more litter and plant cover and less bare ground than the urban site. Graminoids were the dominant vegetation at the exurban plots. In contrast, mostly introduced forbs were found on the urban prairie dog colony. However, the forage quality and quantity tests demonstrated no difference between the two colonies. The relative prairie dog density was greater at the urban colony, which has the potential to drive greater vegetation utilization and reduced cover. Exurban rangeland showed lower levels of impact and retained all of the plant functional groups both on- and off-colony. These results suggest that activities of prairie dogs might further exacerbate the impacts of humans in fragmented urban rangeland habitats. Greater understanding of the drivers of these impacts and the spatial scales at which they occur are likely to prove valuable in the management and conservation of rangelands in and around urban areas.

Corresponding author
Graeme Shannon,
graeme.shannon@colostate.edu

## INTRODUCTION

The North American Great Plains region is a large dynamic ecosystem that is inhabited by a diverse variety of plants and animals, which have generated a heterogeneous landscape made up of three major prairie types—shortgrass, mixed, and tallgrass

(*Lauenroth, Burke & Gutmann, 1999*). Since European settlement rapidly expanded west during the mid-1800s, large portions of the Great Plains ecosystem have undergone dramatic transformation as a function of human population growth driving agricultural and urban development (*Samson & Knopf, 1994*). These habitats continue to face increasing anthropogenic pressure, with the metropolitan areas of the western United States currently experiencing the greatest rate of growth in the country (*Maestas, Knight & Gilgert, 2003*), forcing many of those cities to further develop open spaces within their city limits. Moreover, increased income, mobility and desirability for rural living has led to the conversion of farm and ranch lands to low-density exurban (rural residential) development (*Maestas, Knight & Gilgert, 2003*). For example, exurban population growth for the state of Montana from 1980 to 2000 was estimated to be 143% (*Theobold, 2005*).

Rangelands (including prairies) provide a wide range of ecosystem goods and services, which are strongly influenced by the structure, and dynamics of the vegetation (*Havstad et al., 2007*). Often, vegetation structure and dynamics are described using plant functional groups (i.e., plants with morphological and perhaps physiological similarities; *Pokorny et al., 2005*; *Peters et al., 2006*). Remaining prairie habitats located within the boundaries of urban areas and among exurban development face potentially negative impacts associated with land use change and human population growth in surrounding areas. In addition to direct habitat loss and fragmentation, the native plant communities of these habitats can be substantially altered as a result of non-native plant species being introduced (*Mack et al., 2000*). These introductions also contribute to a loss of biodiversity within the rangelands as a result of native species facing competitive exclusion (*Maestas, Knight & Gilgert, 2003*). These impacts can also extend up the food web, degrading habitat and forage quality for a variety of native wildlife species. One such species that is experiencing severe pressure from development and anthropogenic disturbance is the black-tailed prairie dog (*Cynomys ludovicianus*), which has faced widespread decline across its historic range (*Miller, Ceballos & Reading, 1994*). The decline has been driven by habitat loss, poisoning programs and disease outbreaks (*Miller et al., 2007*). Remaining prairie dog colonies commonly occur in isolated pockets scattered throughout their original range, with many of the more dense colonies found in exurban areas surrounding western cities (*Armstrong, Fitzgerald & Meaney, 2011*).

Black-tailed prairie dogs (referred to herein as prairie dogs) are herbivorous, consuming a wide range of available plant material from grasses to prickly pear cactus (*Hoogland, 1995*). They are considered to be a keystone species and ecosystem engineers because of their ability to alter the landscape and generate refuges and foraging opportunities for an array of species (*Whicker & Detling, 1988*; *Kotliar et al., 2006*). For example, in a functional prairie ecosystem, the foraging and burrowing behavior of prairie dogs has been demonstrated to increase biotic diversity (*Augustine & Baker, 2013*) and influence community structure (*Van Nimwegen, Kretzer & Cully, 2008*) in close proximity to the colony, while also playing an important role in ecosystem function (*Martinez-Estévez et al., 2013*). Nevertheless, prairie dogs are also politically controversial. In agricultural areas, prairie dogs are considered to compete directly with livestock for available forage

(*Vermeire et al., 2004*; *Derner, Detling & Antolin, 2006*), while colonies in fragmented urban habitats have the potential to negatively impact the ecosystem (e.g., loss of native species, soils and litter) due to elevated densities and restricted movement (*Beals et al., 2014*). There are also public health concerns surrounding the transmission of zoonotic diseases such as the plague (*Lowell et al., 2005*). A number of contentious population control measures have therefore been put in place to reduce prairie dog numbers in areas where they are considered to be a nuisance (*Hoogland, 1995*).

The aim of our study was to compare the vegetation of prairie dog habitats in an urban and an exurban rangeland, to explore whether the associated difference in human disturbance may lead to differences in vegetation abundance and composition on and off prairie dog colonies. The predictions were: (1) The presence of a prairie dog colony would reduce the abundance of vegetation and litter and increase the amount of bare ground at both sites, as a function of prairie dog foraging behavior and burrowing activity. (2) The exurban site would support a greater abundance of graminoids (grasses and grass-like plants) and forbs, and have greater quantities of litter and less bare ground than urban site, due to less human disturbance enabling prairie dog foraging and burrowing to be distributed over a larger area. (3) The urban site would have lower native plant cover due to a greater probability of non-native plant species being introduced from nearby developments. (4) The quantity and quality of forage at the exurban prairie dog colony would be greater than at the urban prairie dog colony. (5) Prairie dog density would be greater at the urban colony compared with the exurban colony due to the limited available range and lack of habitat connectivity for animals to disperse to nearby rangelands.

## METHODS

### Study sites

The research was conducted in Fort Collins, Colorado from June to August 2013. Two rangeland study sites (urban and exurban) were selected according to their location relative to the city, proximity to infrastructure and in conjunction with the definitions of urban and exurban developments given in *Theobold (2005)*. Colina Mariposa Natural Area was chosen as the urban site; this rangeland is located in southwestern Fort Collins at the intersection of two busy roads, with urban neighborhoods on the eastern and western borders (Fig. 1). A railroad track bisects the natural area, with the prairie dog colony predominantly located on the east side of the tracks. Pineridge Natural Area was selected as the exurban site, being located on the western edge of Fort Collins in a public open lands district (Fig. 1). It has a reservoir on the northern side, a road to the northwest, while the western boundary is predominantly natural habitat with sparse houses. The southern edge and part of the southeastern side adjoins more open space and a community park, while a small section of the eastern border consists of low-density housing. Both sites were further evaluated to ensure that they had similar topographical characteristics, represented similar rangeland ecological sites and that there was sufficient area to collect data on vegetation abundance and composition, both on and off prairie dog colonies. Fort Collins Natural

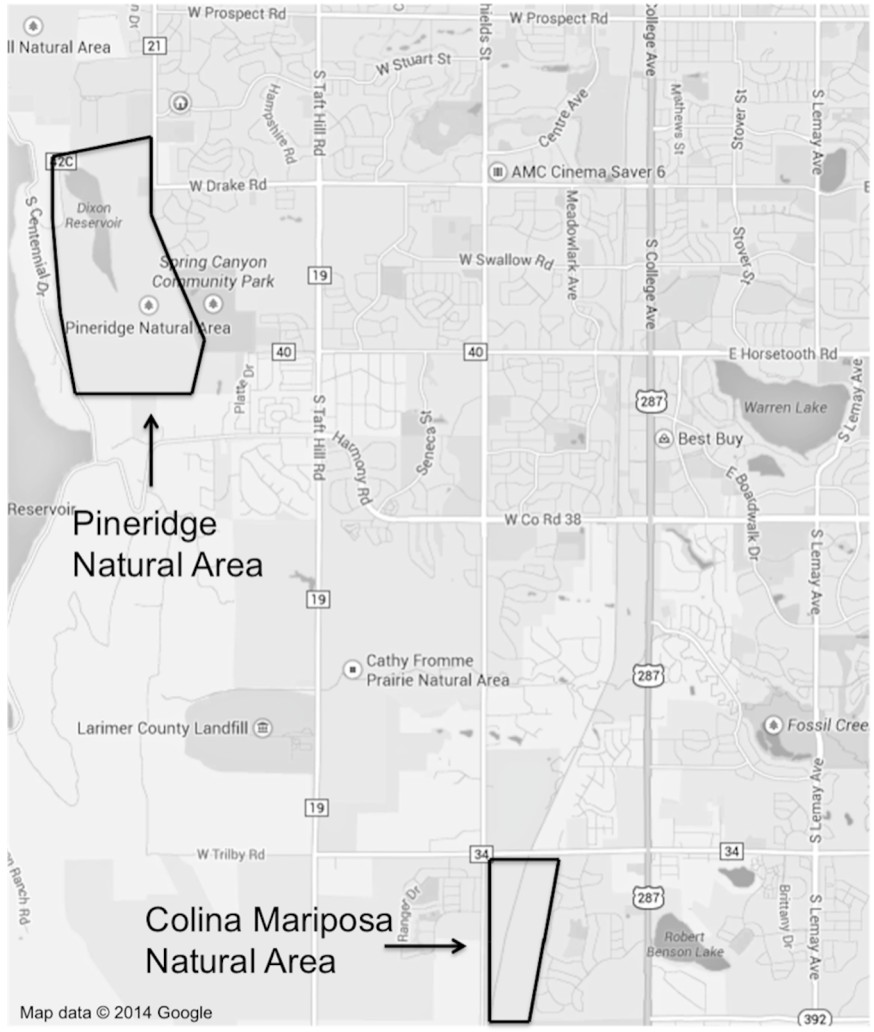

**Figure 1 A map of Fort Collins with the locations and extent of the two study sites outlined.** The latitude and longitude are given below for the center point of the Pineridge Natural Area transects (on-colony: 40.542850, −105.139589 and off-colony: 40.544677, −105.141220) and the Colina Mariposa Natural Area transects (on-colony: 40.483427, −105.093809 and off-colony: 40.485108, −105.091277). Map data © 2014 Google.

Areas provided us with a research permit (#: 296-2012) that stipulated their approval of the proposed study and the conditions under which we could conduct the work.

The vegetation composition varied across the sampling areas (Table 1). Overall, forbs were more common on-colony and graminoids were more common off-colony. The only species that was common to multiple sampling areas was western wheatgrass (*Elymus smithii* (Rydb.) Gould). The species observed on- and off-colony at the urban site were different and there was no overlap in dominants. There was more similarity in the vegetation on- and off-colony at the exurban site with two common dominant species; western wheatgrass and Japanese brome (*Bromus japonicus* Thunb.). There were more introduced species observed at the exurban site than at the urban site.

**Table 1** **Dominant species based on observations along transects established in each sampling area (urban and exurban sites both on and off the prairie dog colonies).** Dominant species were considered as those which occurred in 9 or more of 21, 0.5 m$^2$ quadrats in each sampling area.

| Sampling area | Species | Common name | Growth form | Duration | Growing season | Origin |
|---|---|---|---|---|---|---|
| Urban Off-colony | *Elymus smithii* (Rydb.) Gould | western wheatgrass | grass | perennial | cool | native |
| | *Hesperostipa comata* (Trin. &Rupr.) Barkworth | needle and thread | grass | perennial | cool | native |
| | *Carex filifolia* Nutt. | threadleaf sedge | grass-like | perennial | cool | native |
| | *Eriogonum annuum* Nutt. | annual buckwheat | forb | biennial | cool | native |
| | *Artemisia dracunculus* L. | tarragon | shrub | perennial | warm | native |
| Urban On-Colony | *Convolvulus arvensis* L. | field bindweed | forb | perennial | cool | introduced |
| | *Chenopodium incanum* (S Watson) A Heller | mealy goosefoot | forb | annual | warm | native |
| | *Dyssodia papposa* (Vent.) Hitchc. | fetid marigold | forb | annual | warm | native |
| Exurban Off-Colony | *Elymus smithii* (Rydb.) Gould | western wheatgrass | grass | perennial | cool | native |
| | *Poa pratensis* L. | Kentucky bluegrass | grass | perennial | cool | introduced |
| | *Bromus japonicus* Thunb. | Japanese Brome | grass | annual | cool | introduced |
| | *Psoralidium tenuiflorum* (Pursh) Rdydb. | slimflower scurfpea | forb | perennial | warm | native |
| Exurban On-Colony | *Elymus smithii* (Rydb.) Gould | western wheatgrass | grass | perennial | cool | native |
| | *Bromus japonicus* Thunb. | Japanese Brome | grass | annual | cool | introduced |
| | *Convolvulus arvensis* L. | field bindweed | forb | perennial | cool | introduced |
| | *Linaria dalmatica* (L.) Mill. | dalmation toadflax | forb | perennial | cool | introduced |

## Data collection

Data were collected on and off of the prairie dog colonies at each of the two study sites. On-colony sampling was conducted by locating the approximate center of the colony using prairie dog burrow distribution and animal density as indicators. Once the center point was identified, a thirty-five meter transect was positioned across the colony, with the center point of the transect corresponding to the center of the colony. Two more thirty-five meter transects were established; one on each side of the central transect, running parallel and separated by a distance of 15 m. Selection of the off-colony sampling areas was dependent on the absence of burrows and a minimum buffer of 20 m from the nearest observed evidence of prairie dog activity (e.g., burrow, trail). Once a suitable area was demarcated, three thirty-five meter transects were laid out following the same approach used for the on-colony plots. The methods described above to locate transects in the sampling areas resulted in thorough coverage of the prairie dog colonies and similar sized neighboring off-colony areas. The data used in the analyses are available in Data S1.

## Cover estimates

Canopy cover and vegetation composition were determined using an extended Daubenmire frame (see *Bonham, Mergen & Montoya, 2004*) placed every 5 m along each transect for a total of 7, 0.5 m$^2$ frame locations (subsamples) along each transect (transect = observation). Canopy cover was estimated once each month during the summer (June–August) to track changes in vegetation composition through time. The three data

collection periods were classified as early summer (June), peak standing crop (July) and late summer (August). Cover was categorized into four distinct functional groups: (1) graminoids, (2) forbs, (3) litter, and (4) bare ground. In mid-July (peak standing crop), the graminoid and forb functional groups were further subdivided according to their duration (perennial or annual), origin (native or introduced), and growing season (cool or warm).

## Forage quality and quantity

The quality and quantity of forage available to the prairie dogs on the colony was estimated from three exclosure cages (dimensions (W × L × H): 30 cm × 60 cm × 75 cm) made out of 0.5 cm × 0.5 cm hardware cloth over wire panels placed on each colony (urban and exurban). The exclosure cages were positioned 15 m apart on a line that ran through the center of the sampling area perpendicular to the transects, with the first cage located at the mid-point between transects 1 and 2. In August, all herbaceous material in each cage was clipped to ground level, bagged and placed in drying ovens at 55 °C for one week. The dried material was weighed and then sent for laboratory analysis (Servi-Tech Laboratories, Hastings, Nebraska) to determine the percentage of total digestible nutrients, crude protein, acid detergent fiber, neutral detergent fiber, and a relative feed value index.

## Maximum prairie dog counts

A relative measure of prairie dog density was determined using aboveground counts of animals in a demarcated sampling area (100 m$^2$). Five repeat counts were performed at each site from the same observation point, which was approximately 150 m away from the sampling area. The data were collected from August 15 to August 31, with a minimum of 24 h between repeat counts. The counts were conducted between 7:00 am and 11:00 am and lasted for 90 min with the total number of aboveground animals within the marked observation area recorded every 10 min. A standardized settling time of 30 min was initiated prior to data collection, allowing the prairie dogs sufficient time to return to their normal behavior after the disturbance of the observer's arrival (*Shannon et al., 2014*). The maximum number of animals observed across the 10 observations per count period was selected, generating five independent measures of relative density at each site following the approach of *Menkens, Biggins & Anderson (1990)*. Wet, overcast weather resulted in the fifth count from the urban site being dropped from the analysis due to unusually low prairie dog activity (max count = ~5 individuals). As the study involved minimally invasive vegetation sampling and behavioral observation, an institutional review of the research was not required.

## Data analysis

Cover data for each functional group were analyzed using a mixed modeling procedure and repeated measures analysis (season) to test for the effects of site, presence or absence of a prairie dog colony, season, and all possible interactions. Peak standing crop cover data were analyzed separately to test for the effects of site, presence or absence of a prairie dog colony, and all possible interactions. The data were analyzed using analysis of variance in SAS 9.3 (PROC mixed SAS Institute, Cary, NC, USA). Where *F*-tests identified significant effects,

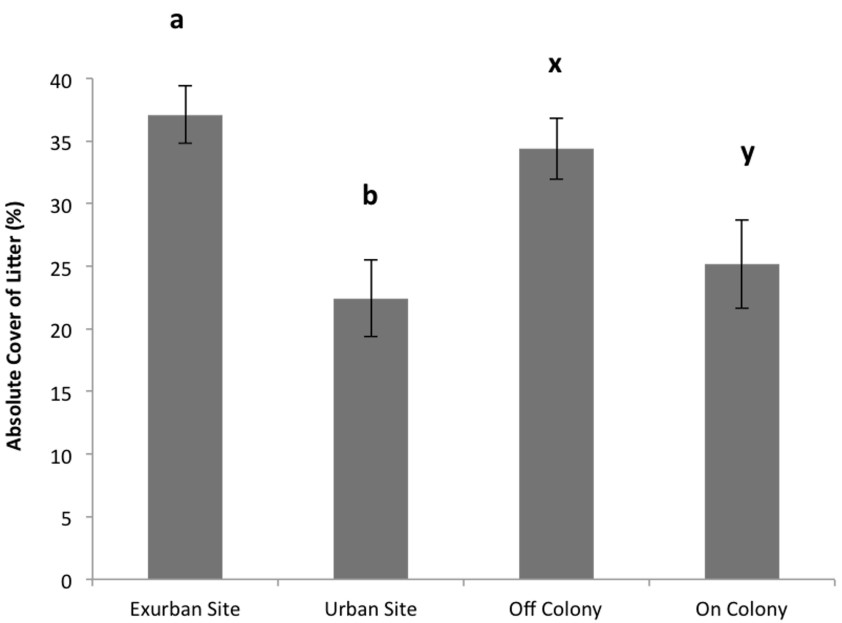

**Figure 2 Absolute cover of litter observed from June–August at the urban and exurban sites both on and off the prairie dog colonies.** Means ($\pm$ S.E.) with the same letter (*a-b* for site and *x-y* for colony) are not significantly different, Fishers LSD $\alpha = 0.05$. Error bars indicate Standard Errors.

the means were separated using Fisher's LSD Alpha $= 0.05$. Data were transformed prior to analysis by calculating the arc-sine square root of the original data to meet assumptions of the analysis. *T*-tests were used to analyze forage quality and quantity and the population density indicator estimates. As the study design involved data collected from two sites, it was not possible to replicate the urban versus exurban comparison in our analyses. All data presented are original scale. A complete table of results from the repeated measures analysis is provided in Table S1.

## RESULTS

### Cover by functional groups

Cover values for the plant functional groups, litter and bare ground were not affected by season ($F = 0.15$–$2.54$, $P = 0.100$–$0.863$), the season by site interaction ($F = 0.04$–$1.37$, $P = 0.272$–$0.96$), the season by colony interaction ($F = 0.16$–$3.34$, $P = 0.053$–$0.855$), or the interaction of all three factors ($F = 0.26$–$2.47$, $P = 0.106$–$0.7706$).

Absolute litter cover was greater at the exurban site than at the urban site ($F = 21.06$, $P < 0.001$) and also greater off-colony than on-colony ($F = 9.11$, $P = 0.006$; Fig. 2). The effect of colony on litter cover was not affected by site ($F = 3.07$, $P = 0.093$). The overall percentage of bare ground was greater at the urban site compared to the exurban site ($F = 48.37$, $P < 0.001$; Fig. 3). There was also more bare ground observed on-colony compared to off-colony ($F = 47.39$, $P < 0.001$; Fig. 3). The effect of colony on cover of bare ground was not affected by site ($F = 2.33$, $P = 0.140$).

Absolute cover of graminoids was affected by site ($F = 94.64$, $P < 0.001$), presence of a prairie dog colony ($F = 135.72$, $P < 0.001$), and the two factors simultaneously ($F = 15.46$,

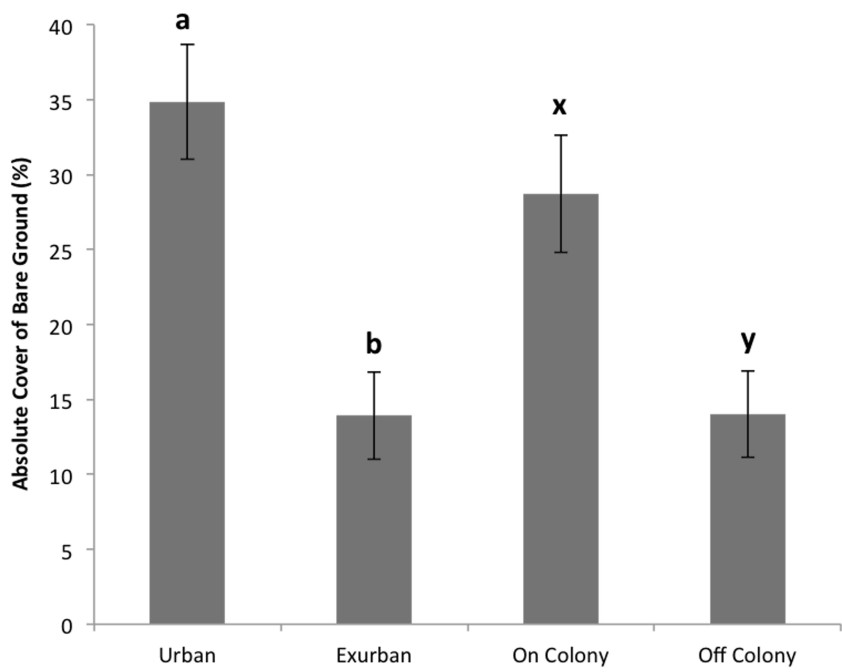

**Figure 3 Absolute cover of bare ground observed from June–August at the urban and exurban sites both on and off the prairie dog colonies.** Means ($\pm$ S.E.) with the same letter (*a-b* for site and *x-y* for colony) are not significantly different, Fishers LSD, $\alpha = 0.05$. Error bars indicate Standard Errors.

$P < 0.0001$). The absolute cover of graminoids ranged from 46% off-colony at the exurban site to 0% on-colony at the urban site, while cover values off-colony at the urban site and on-colony at the exurban site were similar and around 25% (Fig. 4).

Forb cover was not affected by site ($F = 3.39$, $P = 0.078$), but was affected by the presence of a prairie dog colony ($F = 63.04$, $P < 0.001$), and the two factors simultaneously ($F = 26.03$, $P < 0.001$). Absolute forb cover was greatest (40%) on-colony at the urban site and least (5%) off-colony at the urban site (Fig. 5). Absolute cover values of forbs on- and off-colony at the exurban site were similar to one another (18% and 11% respectively), less than the value at the on-colony urban site and greater than the value at the off-colony urban site (Fig. 5).

## Cover at peak standing crop

Perennial native warm season graminoids were not affected by site ($F = 0.79$, $P = 0.399$), but had lower cover values on prairie dog colonies compared to off ($F = 15.43$, $P = 0.004$). The interaction between colony and site was not significant ($F = 0.11$, $P = 0.7525$). Cover of perennial native cool season graminoids was affected by a significant interaction between site and presence of a prairie dog colony ($F = 51.90$, $P < 0.001$; Fig. 6), a result that was driven by the fact that no graminoids were observed throughout the entire growing season on-colony at the urban site (see Fig. 4). However, there were no differences in the abundance of perennial native cool season graminoids among the other sampling areas (Fig. 6).

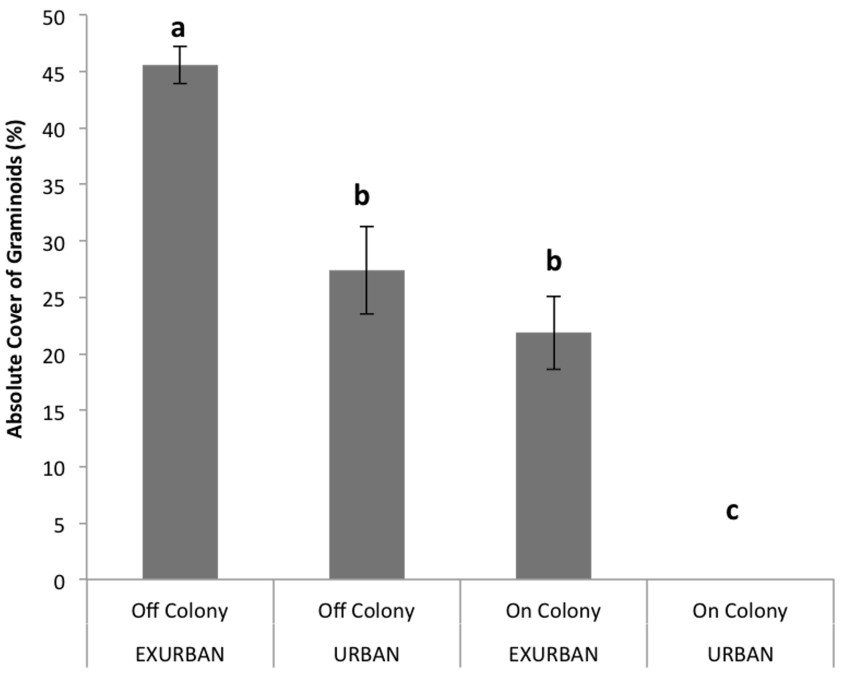

**Figure 4 Absolute cover of graminoid species observed from June–August at the urban and exurban sites both on and off the prairie dog colonies.** Means (± S.E.) with the same letter are not significantly different, Fishers LSD, $\alpha = 0.05$. Error bars indicate Standard Errors.

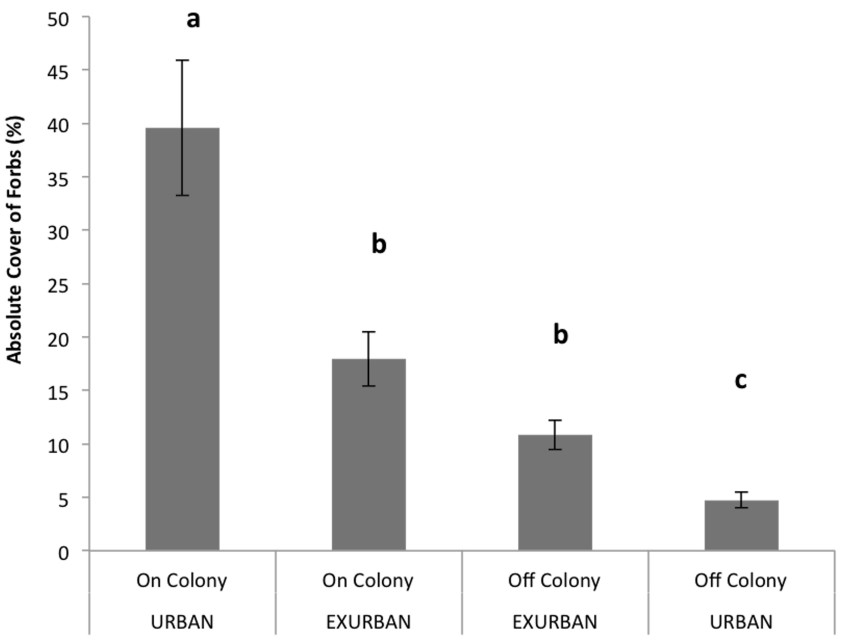

**Figure 5 Absolute cover of forb species observed from June–August at the urban and exurban sites both on and off the prairie dog colonies.** Means (± S.E.) with the same letter are not significantly different, Fishers LSD, $\alpha = 0.05$. Error bars indicate Standard Errors.

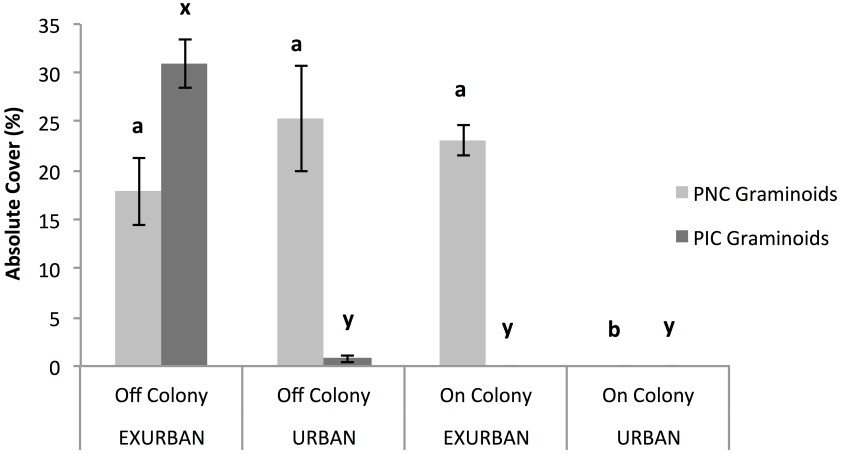

**Figure 6 Absolute cover of perennial native cool season (PNC) and perennial introduced cool season (PIC) graminoids.** Observations were conducted in mid July at peak standing crop at the urban and exurban sites both on and off the prairie dog colonies. Means (± S.E.) with the same letter (*a* and *b* for PNC graminoids; *x* and *y* for PIC graminoids) are not significantly different, Fishers LSD, $\alpha = 0.05$. Error bars indicate Standard Errors.

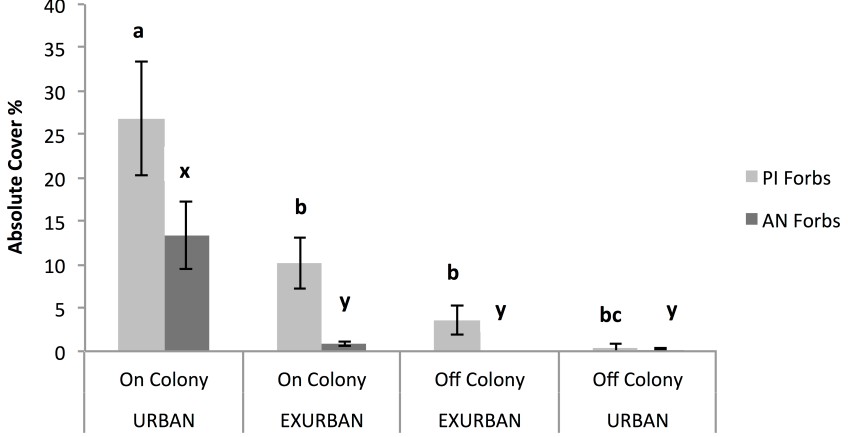

**Figure 7 Absolute cover of perennial introduced forbs (PI Forbs) and annual native forbs (AN Forbs).** Observations were conducted in mid-July at peak standing crop at the urban and exurban sites both on and off the prairie dog colonies. Means (± S.E.) with the same letter (*a* through *c* for PI Forbs; *x* and *y* for AN Forbs) are not different, Fishers LSD, $\alpha = 0.05$. Error bars indicate Standard Errors.

Annual introduced cool season graminoids were only found at the exurban site ($F = 10.32$, $P = 0.012$), while the presence of a prairie dog colony did not affect cover ($F = 0.01$, $P = 0.909$) nor did the two factors interact ($F = 0.2511$, $P = 0.6298$). Perennial introduced cool season graminoids were affected by the two-way interaction between site and colony ($F = 131.29$, $P < 0.001$; Fig. 6). The greatest percent cover by perennial introduced cool season graminoids was observed off-colony at the exurban site (31%), which was mostly *Poa pratensis* L. (see Table 1). At all of the other plots, there was little to no perennial introduced cool season graminoid cover (Fig. 6).

**Table 2 A comparison of forage quantity and quality using *T*-tests.** Data were collected in August from the three exclosure cages located on both exurban and urban prairie dog colonies (see 'Methods').

| Feed test results | Urban (mean) | Exurban (mean) | *T*-statistic | *P*-value |
|---|---|---|---|---|
| Biomass (grams/m$^2$) | 139.72 | 174.33 | −0.520 | 0.642 |
| Digestible nutrients (%) | 63.0 | 64.5 | −0.334 | 0.755 |
| Crude protein (%) | 11.53 | 9.40 | 1.250 | 0.320 |
| Acid detergent fiber (%) | 35.40 | 34.07 | 0.336 | 0.755 |
| Neutral detergent fiber (%) | 45.20 | 56.23 | −2.431 | 0.075 |
| Relative feed value | 127.67 | 104.33 | 1.497 | 0.213 |

Perennial native forb species cover did not vary significantly by site ($F = 3.47$, $P = 0.10$) or the presence or absence of a prairie dog colony ($F = 1.01$, $P = 0.344$). The interaction between site and colony was also non-significant ($F = 0.39$, $P = 0.549$). Bienneial native forbs were only observed off-colony at the urban site.

The abundance of annual native forbs was significantly greater on-colony at the urban site compared with all of the other areas, resulting in a significant two-way interaction between site and presence or absence of a prairie dog colony ($F = 16.80$, $P = 0.003$; Fig. 7). Annual native forbs accounted for 13% cover on-colony at the urban site, reflecting the abundance of two of the three dominant species (see Table 1).

Absolute (%) cover of perennial introduced forbs was not affected by site ($F = 0.49$, $P = 0.502$) but was affected by the presence of a prairie dog colony ($F = 34.73$, $P = 0.001$). The interaction between site and colony was also significant ($F = 11.40$, $P = 0.01$) (Fig. 7). The greatest percent cover by perennial introduced forbs (27%) occurred on-colony at the urban site while cover of this functional group on-colony at the exurban site and off-colony at the urban site were similar. Perennial introduced forb canopy cover was least off-colony at the exurban site. These results reflect the dominance of field bindweed (*Convolvulus arvensis* L.) at three of the four sampled areas (see Table 1). Cover of annual introduced forbs and biennial introduced forbs accounted for ⩽1% across the study areas and therefore values were too small to detect meaningful differences across either site or in the presence or absence of a prairie dog colony.

## On-colony forage

The forage testing analysis and data gathered from weighing the dried biomass revealed no significance differences between sites for the six measures of forage quality and quantity (see Table 2).

## Prairie dog density estimate

The prairie dog population observations taken from each colony indicated that the relative measures of prairie dog densities differed significantly between the exurban and urban colonies ($F = 10.20$, $P = 0.02$). The mean relative density of prairie dogs at the exurban colony was 14 (±2 SE) individuals per hectare, while the density at the urban colony was 19

(±3 SE) individuals per hectare. The dataset comprised nine separate counts, so caution is required with the interpretation of these results.

## DISCUSSION

Our results demonstrated marked differences in vegetation composition between the exurban and urban rangeland sites studied, as well as between areas sampled on and off of a prairie dog colony. The exurban site had more live plant cover and less bare ground compared with the urban site, and the vegetation composition was similar between sampling areas, with the off-colony vegetation predominately comprised of graminoids with fewer forbs, while the on-colony vegetation was a more even mixture of graminoids and forbs. The vegetation composition off-colony at the urban site was a mixture of mostly graminoids with some forbs, while on-colony vegetation was comprised of only forbs with field bindweed the most abundant species, concurring with recent research on prairie dogs in urban habitats (*Magle & Crooks, 2008*; *Beals et al., 2014*). The abundance of bindweed at the urban site is a common feature of disturbed urban and exurban rangeland systems (*Whitson et al., 1998*). Indeed, the success of this plant in colonizing highly disturbed areas suggests that the foraging and burrowing activities of prairie dogs on the urban colony is enabling its propagation (*Magle & Crooks, 2008*; *Beals et al., 2014*). It is important to note at the outset that our study involves the comparison of two distinct rangeland sites, and while these were chosen for their contrasting locations with regard to urban infrastructure and human disturbance, we acknowledge that the lack of site replication limits the inference of our results.

The on-colony data also indicates that prairie dog activity drives the occurrence of bare ground; however, this effect is more pronounced at the urban site. Besides initiating changes in the amount of bare ground, the prairie dog colonies also changed the vegetation structure by decreasing the abundance of graminoids, while increasing forb abundance observed in the community (see also *Magle & Crooks, 2008*; *Beals et al., 2014*). These changes in vegetation structure have also been documented in natural prairie dog habitat and underpin the ecosystem-engineering role of the species (*Whicker & Detling, 1988*; *Baker et al., 2013*). However, if the optimal level of disturbance is exceeded then overgrazing can occur, which has the potential to amplify negative ecological impacts, particularly in prairie habitats that are already impacted by human disturbance (*Beals et al., 2014*). Furthermore, exurban and urban systems are also susceptible to the introduction of non-native species due to their fragmented state and proximity to human activity (*Magle et al., 2010*). There was at least one introduced dominant plant species at all plots, with the exception of the urban off-colony plot where, interestingly, all four dominants were native. The introduced species varied according to plot, species, and growth form, with field bindweed the most commonly observed introduced forb at both on-colony plots. Non-native species are likely to vary in terms of their impacts on prairie habitat, with many species being relatively benign. Bindweed, however, has the potential to be a pervasive weed that can spread rapidly due to a root system which is capable of vegetative reproduction, while the plant produces large numbers of long-lived seeds (*Jacobs, 2007*). This can result

in vast, dense and tangled vine mats that outcompete and ultimately exclude native plant species, particularly in altered or disturbed habitats (*Jacobs, 2007*).

Interestingly, there was no evidence of a significant difference in forage quality or quantity between the urban and exurban site. The abundance of field bindweed at the urban on-colony site is likely to generate significant plant biomass and relatively high values for many of the forage quality measures, rivaling that of the exurban site. Nevertheless, bindweed contains tropane, a potentially toxic alkaloid that led to high levels of mortality in mice that were fed concentrated diets of bindweed (*Schultheiss et al., 1995*). These findings demonstrate that secondary compounds are also crucial when assessing forage quality. Moreover, a greater sample size, and determinations of forage quality at multiple times throughout the growing season, would need to be collected to increase the accuracy of the analysis on forage quality before firm conclusions can be drawn.

The exurban population density indicator estimated fewer prairie dogs per hectare than at the urban site, which along with the greater size of the exurban site may result in an intermediate disturbance regime, enabling the retention of their role as ecosystem engineers. Indeed the disturbance from prairie dogs foraging and burrowing at the larger exurban site is distributed over a greater area, so impact on vegetation is lessened and allows for greater recovery periods for many of the native plants. Whereas elevated densities and reduced dispersal typical of fragmented and disturbed habitats (*Johnson & Collinge, 2004*; *Magle & Crooks, 2008*) can ultimately lead to overgrazing and a loss in vegetation cover, diversity and richness (*Beals et al., 2014*). However, it is worth noting that relative aboveground prairie dog densities of 5 individuals per hectare were documented at an undisturbed colony 40 km from Fort Collins (*Shannon et al., 2014*), significantly lower than those measured at either of the colonies used in this study. In addition to the elevated prairie dog densities that can impact native vegetation cover and species persistence, habitat fragmentation and human disturbance has the potential to affect prairie dog fitness by altering behavior at both the urban and exurban study sites. The close proximity to human disturbances can increase the amount of time that prairie dogs are vigilant while foraging for food (*Ramirez & Keller, 2010*; *Shannon et al., 2014*).

Native annual forbs and perennial native cool season graminoids are absent on-colony at the urban site. Instead, the on-colony sampling area at the urban site was dominated by field bindweed. These results suggest that the presence of prairie dogs is contributing to the relatively disturbed state of the vegetation, which is further compounded by site history, the proximity of the site to agricultural fields and a suburban neighborhood (*Beals et al., 2014*). Furthermore, the dominant species on-colony at the exurban site is represented by three functional groups compared with only two functional groups at the on-colony urban site, which is consistent with the suggestion that the presence of a prairie dog colony can increase the heterogeneity and biotic diversity in more natural prairie habitats (*Coppock et al., 1983*; *Whicker & Detling, 1988*; *Kotliar et al., 2006*; *Baker et al., 2013*). Our findings indicate that the exurban site may have retained a number of its functions as a grassland ecosystem. For example, the only native dominant plant species of the four observed at the exurban sites is western wheatgrass, a cool season grass that has adapted to a grazing

disturbance dynamic. Nonetheless, it is important to note that unlike the study of the protected area (*Coppock et al., 1983*), the other three of the four dominant species at the exurban colony were introduced species (see Table 1). Furthermore, the relationship between the diversity of forb and graminoid species at the exurban site also remained significantly lower than that of the protected natural prairie in Wind Cave National Park (*Coppock et al., 1983*). The urban colony dominant species were made up of three forb species and included one introduced with two native species.

The results we present are from a comparison of two sites and the limitations on inference could be greatly improved by conducting a larger scale study that selected multiple urban and exurban field sites. Such an approach would provide more data and valuable replicates to aid in establishing concrete patterns in the effects of urbanization on habitat fragmentation and the role that prairie dogs play in these altered rangeland systems. Nevertheless, our study demonstrated a marked difference in vegetation composition and habitat disturbance between an exurban and urban rangeland with prairie dogs present. The exurban site retained greater litter cover and less bare ground than the urban site, and a greater abundance of native cool season graminoids on-colony, while the most abundant plant on-colony at the urban site was an introduced, invasive forb. Habitat disturbance and fragmentation also have implications for prairie dogs, which face a greater risk of extinction, loss of immigration and emigration routes, and reduction in genetic variability. Although prairie dog colonies provide a suite of ecosystem services such as improved quality of forage on their colonies for other herbivores, increased turnover of soil nutrients, and decreased soil compaction (*Martinez-Estévez et al., 2013*), these processes may well be compromised in disturbed and fragmented rangeland habitats. A situation that can result in prairie dog colonies exacerbating the impacts associated with human disturbance and environmental change (*Beals et al., 2014*). Based on our initial results, we recommend that the scale-dependent interactions between prairie dogs and vegetation composition be further researched, particularly with regard to their keystone role (see also *Lomolino & Smith, 2003*; *Magle & Crooks, 2008*; *Beals et al., 2014*). Indeed, a greater understanding of these factors will aid effective conservation and management of prairie dog habitats and the integrity of U.S. rangelands, particularly in the face of expanding urban growth.

## ACKNOWLEDGEMENTS

We are grateful to Jennifer Shanahan and Aran Meyer at Fort Collins Natural Areas for sharing their expertise and providing logistical support. We also acknowledge the assistance of faculty and staff members within the Department of Forest and Rangeland Stewardship, the Department of Fish, Wildlife and Conservation Biology, and the Colorado Natural Heritage Program, all in the Warner College of Natural Resources at Colorado State University. Finally, we appreciate the constructive comments and suggestions by the editor and reviewers that have helped improve the manuscript.

### Funding

The project was funded by an Honors enrichment award given to Rebecca Hopson. The funders had no role in study design, data collection and analysis, decision to publish, or preparation of the manuscript.

### Grant Disclosures

The following grant information was disclosed by the authors:
Rebecca Hopson.

### Competing Interests

The authors declare there are no competing interests.

### Author Contributions

- Rebecca Hopson conceived and designed the experiments, performed the experiments, analyzed the data, wrote the paper, prepared figures and/or tables, reviewed drafts of the paper.
- Paul Meiman conceived and designed the experiments, analyzed the data, wrote the paper, reviewed drafts of the paper.
- Graeme Shannon conceived and designed the experiments, wrote the paper, prepared figures and/or tables, reviewed drafts of the paper.

### Animal Ethics

The following information was supplied relating to ethical approvals (i.e., approving body and any reference numbers):

As the study involved minimally invasive vegetation sampling and behavioural observation, an institutional review of the research was not required.

### Field Study Permissions

The following information was supplied relating to field study approvals (i.e., approving body and any reference numbers):

The Fort Collins Natural Areas Research Permit (#: 296-2012) stipulated the conditions under which we could conduct the research at the two study sites in the city of Fort Collins.

### Supplemental Information

Supplemental information for this article can be found online at http://dx.doi.org/10.7717/peerj.736#supplemental-information.

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
