# Peer review of "Rangeland dynamics: investigating vegetation composition and structure of urban and exurban prairie dog habitat"

_PeerJ, doi:10.7717/peerj.736_

## Round 0.1 · original submission · Major Revisions

Please address the concerns of the two reviewers particularly reviewer one. I agree with their view concerning the use of mixed models to explore potential confounding. Increased circumspection regarding the findings in the light of the potential for confounding and lack of replication is also merited. However, with these revisions, this will be a very useful contribution.

·

Basic reporting

The article appears to meet the PeerJ basic reporting criteria.

Experimental design

The experimental design, associated analyses and discussion appear to be flawed, in that one of the central comparisons within the study (urban v. exurban site) is not replicated, and so the urban/exurban comparison is completely confounded with whatever other pre-existing spatial differences there may be between the locations studied (Hurlbert 1984; Heffner et al. 1996; Millar & Anderson, 2004; Chaves, 2010; Ramage et al. 2012). It is also not clear that the authors have accounted for the spatial proximity of the transects on the prairie dog colonies carried out within sites in their mixed model; again, this is a pseudoreplication issue. One can imagine that the analysis could be considerably improved by mixed-model, or spatial (e.g. Dray et al. 2012), analyses that take into account the nesting of transects within colonies, and the nesting of colonies within sites, although, for inferential statistics, there is no obvious way of getting around the fact that one of the central natural comparisons presented (urban/exurban) is not replicated.

I believe, however, that the results are interesting, and could also make a contribution to the wider literature on this topic through meta-analytical studies, I also accept that true replication made be hard in this system, therefore, I do not recommend against publication, merely that the spatial dependencies within the study should be better modelled, and that the urban/exurban comparison should be re-framed throughout the entire paper to emphasise that it is not a treatment that can be rigorously tested within the current experimental framework of the authors, but *could* be a explanator of the between-site differences observed. All conclusions on this comparison should await a greater number of study sites, or the combination of the results presented here with other experiments via meta-analysis.

If the paper is improved in this way, another problem that should be addressed is the inadequate reporting of model structure and results. Analysis of variance/deviance tables (plenty of space these days in Supplemental Material), or at the very least degrees-of-freedom information, should be reported for each model (more information like this would also likely facilitate meta-analyses within the study area if the raw data are not to be provided). Given the number of tests conducted, it would also be good to acknowledge this by not placing too much weight on marginal P-value results in the Discussion.

References
Dray, S., Pélissier, R., Couteron, P., Fortin, M. J., Legendre, P., Peres-Neto, P. R., ... & Wagner, H. H. (2012). Community ecology in the age of multivariate multiscale spatial analysis. Ecological Monographs, 82(3), 257-275.
Heffner, R. A., Butler, M. J., & Reilly, C. K. (1996). Pseudoreplication revisited. Ecology, 2558-2562.
Hurlbert, S. H. (1984). Pseudoreplication and the design of ecological field experiments. Ecological monographs, 54(2), 187-211.
Millar, R. B., & Anderson, M. J. (2004). Remedies for pseudoreplication. Fisheries Research, 70(2), 397-407.
Chaves, L. F. (2010). An entomologist guide to demystify pseudoreplication: data analysis of field studies with design constraints. Journal of medical entomology, 47(3), 291-298.

Validity of the findings

In the addition to the comments provided in Experimental Design, I also have an issue with the mention of findings related to habitat fragmentation within the abstract and throughout the paper. The way fragmentation is talked about in the abstract makes it seem like the effects of fragmentation are explicitly tested in the paper, whereas it gradually emerges that the issue of fragmentation only applies to assumed levels of differences in within-site fragmentation between the urban and exurban sites, and that this is merely another factor that varies between the two sites, alongside assumed human disturbance levels, and other issues that aren't considered that may affect between-site differences (e.g. historical disturbances/site management). This issue is again related to the fact that the paper does not actually present a replicated test of urban/fragmented v. exurban/less fragmented 'treatments'.

Line 307: The Prairie Dog Density Estimate is significant at 0.02 using a t-test (line 217), but this does not take into account the nesting of colonies within sites (and it is not clear how the repeated counts were accounted for, presumably averages were taken, which wastes information), so it seems unlikely that there is going to be a significant difference of dog density between the sites. This is related to the points in Experimental Design.

Finally, there is no mention of how the data are to be archived or made available. PeerJ reviewing criteria state 'The data on which the conclusions are based must be provided or made available in an acceptable discipline-specific repository.'

Additional comments

The paper is in general otherwise well-written and presented, and only a few very minor comments follow:

Line 33: 'Grasses and grass-likes' typically called graminoids in most papers I have seen, but perhaps this is a rangeland-specific/regional usage.
Line 50: 'likely to' for 'will' more appropriate?
Line 135: typo, 'a' not needed
Line 214: 'Data were transformed', this is not generally recommended nowadays, e.g. Warton & Hui (2011).
Line 227 (and other places) use of hyphens for on-/off-colony inconsistent.
Line 285 dominance for 'common dominance', which sounds like a new piece of terminology and is confusing.
Line 299: Bromus japonicus mis-spelling.
Line 375: 'is' for 'are'

# Fig 1.
The authors should give GPS references for the study sites (e.g. centres of prarie dog colonies with GPS error?, or ensure these are deposited with the raw data) or give x,y coordinates on the map provided.

# Paper references
Line 468: Magle et al. missing paper title
Line 488: Shannon et al. missing paper title
Line 489: Theobold. paper missing part of title

Reference
Warton, D. I., & Hui, F. K. (2011). The arcsine is asinine: the analysis of proportions in ecology. Ecology, 92(1), 3-10.

·

Basic reporting

General :
Be consistent writing off-colony or off colony. Same for on-colony/ on colony.

Introduction
L99: briefly state the diet of prairie dogs
L91-103: you have stated the role of prairie dogs in the ecosystem, is there evidence of negative impacts on the ecosystem too?
General: it would be good to know why the prairies are important (ecosystem services)/why they should be conserved. What comprises a prairie in good condition – what species? You have investigated cool/warm season grass; native/introduced species; annual/perennial with little indication as to why these factors might be important.

Experimental design

Method
L186-187: It is not clear where the cage located at 7.5m into the plot area is located. 7.5m from where? (I wonder why these measures were not carried out on the ‘off colony’ sites too).
L195: Not clear when the 5 repeat counts of prairie dogs were carried out.

Validity of the findings

Results
Generally – very thorough
L221: Avoid use of first person (we)
L222-224: Don’t cite values where no significance.
L228: I don’t understand fully ‘the effect of colony on litter cover was consistent at the two sites’ given that you have just stated that ‘litter cover was … greater off-colony than on colony’. Same at L232. Think I know what you mean, but not clear.
General: I suggest you include degrees freedom in stats. Only state if P<0.05; P<0.01 or P<0.001) Don’t provide actual P values
L291: Table 1 is fine but in terms of species occurrence, tells us very little. Please also present the data to provide information about relative dominace e.g. in L174 you state that you counted the number of times each of dominant species occurred – please present this information. (This will help the discussion.)
Figs 2 & 3: graphs should be altered because the same data appear twice on the same axis. Although the y-axis is the same, I reckon splitting the graph into two (though keeping them next to each other) would be preferable.
Discussion
L317: compare ‘with’ not compare ‘to’. (Pedantic – but preferable.)
L319: ‘comprising grassses’. Comprised should not be followed by ‘of’.
L355-356: change to ‘urban site could reduce their movement and dispersal’ – so it is clear that it is the dogs who are moving and dispersing
L356: don’t like the grammar of the sentence ‘processes which appear to be impacted to a lesser extent t the exurban site’ – How can processes be impacted? (‘which’ should be ‘that’). Can’t suggest alternative because not sure what is meant.
L364-372: not clear how this relates to the findings
L369: in UK we say ‘acclimatized’ not ‘acclimated’ - maybe both are therefore acceptable. (Also elsewhere: UK spelling of fiber is fibre. - Both are probably acceptable)
L374: not clear what you mean by ‘key plant functional groups are absent from the urban site’. I assumed that the ‘grasses and grass-likes’ were one functional group and the ‘forbs’ were another. These were then split further into categories – but you say they were absent from the urban site; or at least the ‘key’ ones were. Which ones are ‘key’? L399-400 also states that ‘the exurban site retained a greater number of plant functional groups..’ – and not only am I not sure what you mean by functional groups, but I’m not convinced that this information is evident from the results. As far as I can see, you first introduce the term ‘functional group’ at line 206; maybe you should clarify what you mean.
L380: You have stated that the prairie dogs retain their role as ecosystem engineers at the exurban site because of the lower density, but you have not explained why you think this to be the case. What ‘engineering’ do you perceive they doing at this site that they aren’t doing at the other sites? You state that there was a greater diversity in dominant species here (L381); I assume this is an attempt to answer this question, but where is the evidence that there is a greater diversity in dominant species at this site? You are discussing diversity of species (also L392), but you have not got any actual diversity measures. You have not identified the other species present, so you do not know which is the most diverse site. Are there ‘indicator species’ in this plant community that would suggest whether it is in ‘good’ or ‘poor’ condition? Do the species that you have identified suggest anything about condition?
L400: you state that ‘the urban on-colony plot was dominated by a single introduced species and bare ground’ – this information is not presented in the results. The results state ‘dominance’ of Chenopodium incanum, Dyssodia papposa and Elymus smithii are all present here.
L403-406: I would have liked to have read this in the introduction-around about L91. Also, if prairie dogs are considered to have a ‘keystone’ role, I would have like to have read about that in the introduction too.
General: No repeat sites. Analysis relies on pseudoreplication. Although the final sentence states that more sites are required, there is no acknowledgement of the lack of replication in this site and of the limitation of the interpretation of the results.
General: the discussion often does not attempt to explain the results or discuss their significance, quite often it merely re-states them. So for instance, in L334 – it is evident that the dogs decreased the abundance of grasses. Is this beneficial or harmful? You have stated that field bindweed is dominant (L344), in the introduction (L78) you have suggested that non-native species would be a problem, but you have not expanded on this in the discussion. Are all non-native species troublesome?

Additional comments

Summary
An interesting paper, mostly very clearly and concisely written therefore it is very easy to understand the methods. The analysis is thorough. Discussion needs to aim to explain findings in relation to literature more; more should be done in terms of reporting the plant species data and discussing their significance.

---

## Round 0.2 · accepted · Accept

Thank you for responding positively to the suggestions of the reviewers and for increasing the clarity of your methods. I hope you will consider publishing further work in peerJ in the future.

·

Basic reporting

I believe that the article now meets all of the PeerJ reporting criteria.

Experimental design

The thorough and well-documented responses by the authors to the concerns of both reviewers have satisified me that the experimental design and its documentation in the manuscript now meet the PeerJ requirements. The recommend to the authors that they make their peer review responses available to increase the value of the article.

Validity of the findings

I believe that the findings are now appropriately caveated through the study in line with the concerns of both of the reviewers.

Additional comments

I was very impressed with the quality and thoroughness of the authors' responses to the reviewers concerns.

# Minor typos comments etc on this version
# Line numbers refer to the uploaded doc with amendments, not the reviewing MS
L114-117 'and' missing from L115, i.e. 'and to explore'
L202: spaces between '1&2'
L221 & L519-520: Reference inconsistency and year missing from reference in L519
L246-326: I'm not sure if it is PeerJ style to have the test stats and P value numbers in italics or not, but I find having all the numbers in italics harder to read.